# *ERG3*-Encoding Sterol C5,6-DESATURASE in *Candida albicans* Is Required for Virulence in an Enterically Infected Invasive Candidiasis Mouse Model

**DOI:** 10.3390/pathogens10010023

**Published:** 2020-12-31

**Authors:** Tatsuro Hirayama, Taiga Miyazaki, Makoto Sumiyoshi, Nobuyuki Ashizawa, Takahiro Takazono, Kazuko Yamamoto, Yoshifumi Imamura, Koichi Izumikawa, Katsunori Yanagihara, Shigeru Kohno, Hiroshi Mukae

**Affiliations:** 1Department of Respiratory Medicine, Nagasaki University Hospital, Nagasaki 852-8501, Japan; tatsuro_h_20@nagasaki-u.ac.jp (T.H.); maksumiyoshi@gmail.com (M.S.); nashizawa-ngs@umin.ac.jp (N.A.); takahiro-takazono@nagasaki-u.ac.jp (T.T.); kazukomd@nagasaki-u.ac.jp (K.Y.); yimamura@nagasaki-u.ac.jp (Y.I.); s-kohno@nagasaki-u.ac.jp (S.K.); hmukae@nagasaki-u.ac.jp (H.M.); 2Department of Infectious Diseases, Nagasaki University Graduate School of Biomedical Sciences, Nagasaki 852-8501, Japan; koizumik@nagasaki-u.ac.jp; 3Department of Laboratory Medicine, Nagasaki University Hospital, Nagasaki 852-8501, Japan; k-yanagi@nagasaki-u.ac.jp

**Keywords:** *Candida albicans*, *ERG3*, ergosterol, pathogenesis, colonization, intestinal tract

## Abstract

Gastrointestinal colonization by *Candida* species is considered the main source of candidemia. The *ERG3* gene in *Candida albicans* encodes a sterol C5,6-desaturase, which is essential for ergosterol biosynthesis. Although *ERG3* inactivation shows reduced virulence in mouse models of disseminated candidiasis, the role of *ERG3* in intestinal infections is unknown. Here, we infected mice with the *C. albicans* strains CAE3DU3 and CAF2-1, containing mutant and wild-type *ERG3*, respectively, and studied gut infection and colonization by these strains. We found that the CAE3DU3 strain showed reduced colonization, pathogenesis, damage to gut mucosa, and chemokine production in the mouse model of invasive candidiasis. Additionally, mice inoculated with CAE3DU3 showed lower mortality than mice inoculated with CAF2-1 (*p* < 0.0001). Chemokines were less induced in the gut inoculated with CAE3DU3 than in the gut inoculated with CAF2-1. Histopathologically, although the wild-type gene was associated with a higher pathogenicity and invasion of the gut mucosa and liver tissues causing remarkable tissue necrosis, the *erg3*/*erg3* mutant was associated with a higher accumulation of cells and lower damage to surrounding tissues than wild-type *ERG3*. These results establish that the ergosterol biosynthetic pathway may be associated with *C. albicans* gut colonization and subsequent dissemination.

## 1. Introduction

Candidemia is cited as the fourth leading cause of bloodstream infections [1]. *Candida albicans* is the most frequently isolated species that is associated with candidemia [2]. Several species of *Candida* are ubiquitous commensal yeasts and are normal components of human skin and gut microbiota, with estimates of *C. albicans* carriage in healthy individuals ranging from 30–60% [3]. Colonization is considered as a prerequisite for developing infection, and the gut is an important site of growth for *Candida* in the development of candidemia [4]. When the intestinal mucosal barrier is damaged (for example, by anticancer chemotherapy), the inhabiting *C. albicans* can further damage the intestinal mucosal barrier, which allows direct invasion of *C. albicans* into the bloodstream and abdominal cavity. Impairment of the host immune response allows the overgrowth of *C. albicans* and dissemination into the bloodstream and adjacent organs, leading to the establishment of infections in various organs [5].

Azole antifungal agents inhibit the biosynthesis of ergosterol, the major sterol of cell membrane. The *ERG3* gene in *C. albicans* encodes a sterol C5,6-desaturase, which is essential for ergosterol biosynthesis. Altered ergosterol synthetic pathways, owing to the inactivation of sterol C5,6-desaturase, are known factors that contribute to azole resistance in vitro [6,7]. The role of *ERG3* in fungal virulence remains controversial. Although *ERG3* inactivation results in reduced virulence in mouse models of disseminated candidiasis and certain clinical isolates harboring mutations in *ERG3* show defects in hyphal growth and virulence [8,9], a clinical isolate that harbors an *ERG3* mutation but displays wild-type hyphal production and virulence has also been reported [10]. The contributions of *ERG3* to colonization and pathogenesis in the gut are not presently known. In this study, we assessed the virulence of *C. albicans erg3* null mutant in a mouse model of invasive candidiasis caused by fungal translocation from the gut.

## 2. Results

### 2.1. Loss of ERG3 in C. albicans Results in Attenuated Virulence in an Enterically Infected Invasive Candidiasis Mouse Model

We assessed the virulence of the *C. albicans erg3* null mutant CAE3DU3 (*erg3*/*erg3 ura3*/*URA3*) and control strain CAF2-1 (*ERG3*/*ERG3 ura3*/*URA3*) in mice, which were inoculated intragastrically. All mice were also treated with cyclophosphamide, thereby resulting in 100% mortality during the period of this experiment. The mice inoculated with CAF2-1 showed 100% mortality by day 11 post-inoculation. In contrast, the mice inoculated with CAE3DU3 survived 11 days post-inoculation and showed 100% mortality by day 19 post-inoculation. The mice inoculated with CAE3DU3 showed lower mortality rates compared to that of mice inoculated with CAF2-1 (*p* < 0.001) (Figure 1).

### 2.2. C. albicans erg3 Null Mutant Has a Lower Colonization and Dissemination Capacity in the Gut

Colonization in the intestinal tract was expressed as log_10_ colony-forming unit (CFU)/mg in the stools (Figure 2). In this experiment, cyclophosphamide was not administered to allow all the mice to survive until 21 d post-inoculation in order to investigate the colonization capacity of CAF2-1 and CAE3DU3 for a longer period. Stool specimens were collected from groups of 8 live mice at 7 d, 14 d, and 21 d post-inoculation. The number of CFUs of CAEDU3 in the stools was lower than that of CAF2-1, with significant differences (d 7, *p* = 0.0003; d 14, *p* = 0.002; d 21, *p* = 0.0002).

To evaluate the subsequent dissemination from the intestinal tract, the liver and kidney fungal burden were examined at 9 d or 14 d post-inoculation (Figure 3). The mice inoculated with CAF2-1 were euthanized 9 days post-inoculation because they could not survive for 14 d post-inoculation (Figure 1). As shown in Figure 3, *Candida* cells were isolated from the livers of all mice inoculated with CAF2-1 (8/8) at 9 d post-inoculation. In contrast, no *Candida* cells were isolated from mice inoculated with CAEDU3. The fungal burden of CAEDU3 was significantly less than that of CAF2-1 in the liver (*p* = 0.0002). In the kidneys, the isolation ratio was 50% for mice inoculated with CAF2-1 (4/8). No *Candida* cells were recovered from mice inoculated with CAEDU3, although the difference in the fungal burden (log_10_ CFU/organ) between the CAF2-1 and CAE3DU3 groups was not statistically significant (*p* = 0.08). At 14 d post-inoculation, *Candida* cells were isolated from the livers and kidneys of all mice inoculated with CAE3DU3 (8/8). *Candida* cells could not be isolated from the blood of any mouse inoculated with CAF2-1 (day 9) and CAE3DU3 (days 9 and 14).

### 2.3. Chemokines Levels in the Gut and Blood of Mice Infected with the erg3 Null Mutant Were Significantly Lower Than Those in Mice Infected with the Wild-Type Strain

To evaluate the inflammation of the gut following *Candida* invasion, we measured the levels of chemokine (C-X-C motif) ligand 1/keratinocytes-derived chemokine (CXCL1/KC) and CXCL2/macrophage inflammatory protein 2-alpha (MIP-2) in the serum and intestinal homogenates using enzyme-linked immunosorbent assay (ELISA) at 9 d post-inoculation. Both CXCL1/KC and CXCL2/MIP-2 levels in the colon tissues and blood were significantly less induced in mice inoculated with CAE3DU3 compared to those in mice inoculated with CAF2-1 (Figure 4).

### 2.4. Histopathological Examination

The small intestine, cecum, colon, liver, and bilateral kidneys of at least three mice inoculated with CAF2-1 and CAE3DU3, respectively, were examined histopathologically (Figure 5). CAF2-1 was present in both hyphal and yeast forms and invaded the intestinal mucosa of the cecum and colon, showing remarkable tissue necrosis at 9 d post-inoculation (Figure 5a). Invasion of CAF2-1 and surrounding tissue necrosis were observed in the livers but not in the kidneys. *C. albicans* cells were observed in the mucosa of the cecum and colon, livers, and kidneys of mice inoculated with CAE3DU3 at 14 d post-inoculation (Figure 5b) and were not identified in any organs at 9 d post-inoculation. *C. albicans* CAE3DU3 was present in both hyphal and yeast forms. However, the invasion and tissue necrosis shown by them was localized compared to those associated with CAF2-1. Cell accumulation was observed in the livers and kidneys without surrounding tissue necrosis.

## 3. Discussion

Ergosterol is a key component of the fungal cell membrane. The ergosterol synthetic pathways can be altered via an *ERG3* mutation that inactivates sterol C5,6-desaturase [7]. The loss of sterol C5,6-desaturase activity is associated with reduced virulence in mouse models of disseminated candidiasis developed by intravenous injection of *Candida* cells [8,9]. The present study revealed that *ERG3* is involved in the gut colonization and subsequent dissemination of *C. albicans*. The analysis of colonization levels in the gut suggested that the *erg3* mutant cells were not eliminated but unable to colonize the gut mucosa at cell concentrations similar to those of the wild-type cells. A possible explanation for the consistent difference in colonization level is that the *erg3* mutant cells do not replicate as well as the wild-type cells do in the gut environment. However, it is difficult to verify this hypothesis. Strains of *C. albicans* containing *erg2*, *erg3*, and *erg24* mutations successfully colonized the mouse vagina [11,12]. On the other hand, strains containing *erg3* and *erg11* mutations failed to colonize the mouse lingual surface, as observed in a murine oropharyngeal candidiasis model [13]. These results suggest that the colonization capacity associated with alterations in the ergosterol biosynthetic pathway may vary depending on the mutations in the *ERG* genes and niches in the hosts.

The reduced virulence of the *erg3* mutant was observed in the survival experiments and histopathological examination. Histopathologically, the CAF2-1 cells were present in both hyphal and yeast forms, invaded the intestinal mucosa, and induced remarkable tissue necrosis. In contrast, the CAE3DU3 cells exhibited a milder invasion and hyphal formation compared with the CAF2-1 cells. The mortality in mice inoculated with CAF2-1 was thought to be a result of intestinal necrosis, since *C. albicans* was found to have invaded the gut mucosa and cause remarkable necrosis. In our previous experiments, although colonization in the gut was detected at 6 d post-inoculation of *C. albicans* wild-type cells (approximately 10^4^ CFU/mg of stool) and no *Candida* cells were found in the intestinal tract histopathologically at 7 d post-inoculation [14]. It is important to explore how invasive candidiasis develops over time, but it seems to be difficult with our mouse model. The mortality in mice inoculated with CAE3DU3 was thought to be a result of disseminated candidiasis, since a large number of *C. albicans* cells was isolated from the liver and kidneys, and intestinal necrosis was not severe at 14 d post-inoculation. Low pathogenicity of the *erg3* mutant may be reflected in the low mortality despite a large number of cells in the organs. The formation of *C. albicans* hyphae is considered the most essential virulence factor [15,16]. Although *erg3* mutants are associated with a deficiency in formation of hyphae in disseminated candidiasis models [8,9,10,17], they can form hyphae in mouse models of oral mucosal [13] and vaginal infections [12]. These studies indicate that *erg3* mutant cells may be able to form hyphae in contact with epithelial cells.

CXCL-1 is involved in the mobilization of leukocyte infiltrates, particularly neutrophils, toward the infection site [18], and is an important mediator in regulating systemic *C. albicans* infection locally [19]. CXCL-2 is rapidly produced at the infection site and recruits phagocytic effector cells into the local infection site where pathogens such as *C. albicans* are present [20]. When strains containing null mutants of *ERG3* were co-cultured with epithelial cells, a reduction in the adhesion, damage, and cytokine production was observed in vitro [13]. In our study, we found that the production of CXCL1/KC and CXCL2/MIP-2 was reduced in both the colon and serum of mice infected with CAE3DU3. The mild inflammation associated with the *erg3* mutant was due to the few chemokines induced in the host mice compared with that associated with the wild-type strain.

Recently, Witchley et al. [21] reported that the yeast-to-hypha morphogenesis programs control the balance between commensal and invasive growth in the gut. Interestingly, commensal fitness was not related to cell shape but inversely related to expression of hypha-specific virulence effectors, such as the transcription factor Ume6 and secreted protease Sap6. In the present study, loss of Erg3 reduced both invasion and colonization in the gut.

A limitation of this study is that only a laboratory-generated *erg3* mutant was examined. Vale-Silva et al. [10] reported that a clinical isolate having an *ERG3* mutation exhibited hyphal production and virulence similar to a wild-type strain. However, as the authors suggested, it is likely that the decreased in vivo fitness due to the *ERG3* mutation was compensated by an unknown mechanism(s) in this particular strain.

In summary, using an enterically infected invasive candidiasis mouse model, the present study demonstrated that the *C. albicans* wild-type cells were present in both yeast and filamentous forms and invaded the gut mucosa, inducing a proinflammatory response and tissue necrosis. The *erg3* mutant was less virulent than the wild-type strain in the survival assay, because loss of *ERG3* reduced the capacity of colonization, invasive growth in the gut, and subsequent dissemination into the adjacent organs. Our results support the fact that, even if *erg3* mutants are highly resistant to azoles in vitro, such clinical isolates have only rarely been reported. Future investigations will be required to elucidate detailed molecular mechanisms linking alterations in the ergosterol biosynthesis pathway and attenuation in the virulence in *C. albicans*.

## 4. Materials and Methods

### 4.1. Candida Strains and Culture Conditions

We used a *C. albicans erg3/erg3* mutant strain CAE3DU3 (*erg3/erg3 ura3/URA3*), and a wild-type *URA3* was inserted back into its native locus in the *erg3* homozygote. The details of strain construction have been described previously [9]. Additionally, we used a control strain CAF2-1 (*ERG3/ERG3 ura3/URA3*) [22], which is similarly a *ura3/URA3* heterozygote. *C. albicans* from frozen stock (−80 °C) was inoculated into yeast peptone dextrose (YPD) medium (1% yeast extract, 2% peptone, and 2% glucose) at 30 °C.

### 4.2. Animals and Ethics Statement

The animals used in this study were specific pathogen-free female DBA2/J mice (5 weeks old; Clea Japan, Inc., Japan). All animals were maintained under specific pathogen-free conditions. All animal experiments were performed in accordance with the Guide for the Care and Use of Laboratory Animals [23], and all institutional regulations and guidelines for animal experimentation following review and approval by the Institutional Animal Care and Use Committee of Nagasaki University (protocol number 1906121536).

### 4.3. Murine Model of Candidiasis from the Gut

A murine model of candidiasis from the gut was established as described previously [14]. Briefly, 5-week-old female DBA2/J mice were fed a low-protein diet with 5% casein (Oriental Yeast Co, Ltd., Tokyo, Japan) for 14 days before inoculation. Sterile water was provided before inoculation, and sterile water containing enrofloxacin (200 mg/L) and vancomycin (500 mg/L) were provided after inoculation. We previously reported that there were no differences in either the degree of fungal colonization, fungal burden in the liver, or survival rates among mice infected with a single 0.2 mL inoculation of *C. albicans* wild-type cell suspension containing 5 × 10^6^ CFU/mL, 5 × 10^7^ CFU/mL, or 5 × 10^8^ CFU/mL [14]. Therefore, the number of *Candida* cells for inoculation was adjusted to 5 × 10^6^ cells/mL. After fasting for 24 h, the mice were inoculated intragastrically with 0.2 mL of cell suspension (1 × 10^6^ CFU/mouse) on day 0 via a stainless-steel catheter. The mice were administered intraperitoneally with 150 mg/kg of cyclophosphamide (Sigma-Aldrich, Japan) on 4 d, 7 d, 10 d, 13 d, and 16 d post-inoculation. Cyclophosphamide was not administered to mice for the colonization assessment to compare fungal levels in the stools.

### 4.4. Isolation of Candida Cells from the Blood, Liver, Kidney, and Stool

The DBA2/J mice inoculated with CAF2-1 and CAE3DU3 were euthanized at 9 d post-inoculation, and those inoculated with CAE3DU3 were euthanized at 14 d post-inoculation (n=8 for each group at each timepoint). All mice were alive before euthanasia in this experiment. Blood was collected from the inferior vena cava after anesthetizing the mice with isoflurane, and 200 µL of each blood sample was directly plated onto YPD agars with imipenem (200 mg/L). The livers and bilateral kidneys were aseptically excised and extensively homogenized. Stools from live mice were collected, weighed, and homogenized on 7 d, 14 d, and 21 d post-inoculation. Homogenates of organ and stool were diluted appropriately and were plated on YPD agars with imipenem (200 mg/L). Colonies were counted after incubating the plates for 24 h at 30 °C. These animal experiments were performed twice on independent occasions to ensure reproducibility.

### 4.5. Chemokine Detection Assay

The mice inoculated with CAF2-1 and CAE3DU3 were euthanized at 9 d post-inoculation. Blood was collected from the inferior vena cava. Large intestines were collected in 1.5 mL tubes containing 500 µL of a solution prepared by dissolving 1 tablet of complete mini protease inhibitor cocktail (Roche, Germany) in 10 mL of phosphate-buffered saline, which was then homogenized. Subsequently, 40 µL of Cell Lysis Buffer M (FUJIFILM Wako Pure Chemical Corporation, Japan) was added to the homogenates, which were allowed to stand at 4 °C for 30 min. The blood samples and homogenates were centrifuged and the supernatants were stored at −30 °C until use. Chemokine levels were measured using CXCL1/KC and CXCL2/MIP-2 ELISA kits (R&D Systems, Minneapolis, MN, USA).

### 4.6. Histopathological Examination

The organs of at least three DBA/2J mice per inoculated strain were examined histopathologically. Mice inoculated with CAF2-1 and CAE3DU3 were euthanized at 9 d post-inoculation and mice inoculated with CAE3DU3 were euthanized at 14 d post-inoculation. Specimens of the small and large intestine, liver, and bilateral kidneys were obtained. No intestinal lavage was performed to avoid excluding intraluminal microorganisms. All tissues were stained with hematoxylin-eosin and periodic acid Schiff.

### 4.7. Statistical Analysis

All statistical analyses were performed using Prism 6.0 software (Graphpad Software Inc., La Jolla, CA, USA). The log rank (Mantel-Cox) test was used for pairwise comparison of survival curves. The differences of fungal burden in the organs and stools, and the chemokine levels were evaluated using the Mann–Whitney U test. A *p* value < 0.05 was considered significant.

## Figures and Tables

**Figure 1 pathogens-10-00023-f001:**
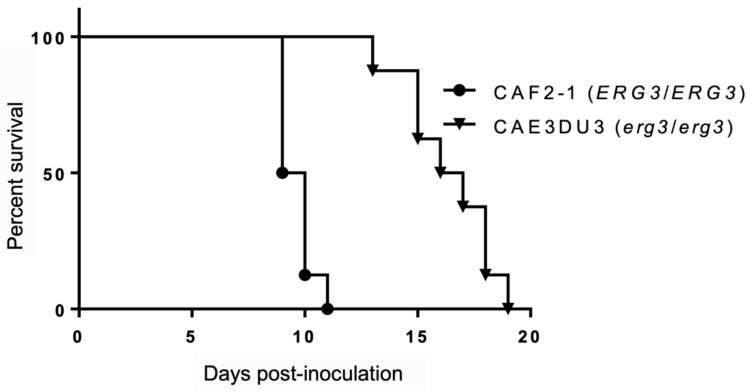
Survival analysis using an enterically-infected invasive candidiasis mouse model. Groups of eight mice were inoculated intragastrically with cell suspensions of CAF2-1 and CAE3DU3 on day 0 of the experiment, and they were treated with cyclophosphamide on 4, 7, 10, 13, and 16 d post-inoculation. Their survival was monitored for 19 d post-inoculation. The mice infected with CAF2-1 exhibited higher mortality rates compared to those of mice infected with CAE3DU3 (*p* < 0.001). Kaplan-Meier curves were created and compared using the log rank (Mantel-Cox) test.

**Figure 2 pathogens-10-00023-f002:**
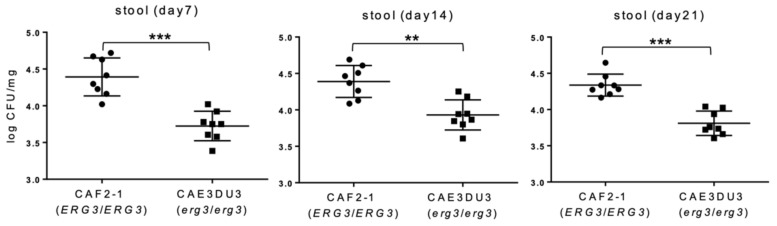
Comparison of colonization in the gut between *C. albicans* strains CAF2-1 and CAE3DU3. Colonization in the intestinal tract was expressed as log_10_ CFU/mg in stool specimens. In this experiment, cyclophosphamide was not administered to allow all the mice to survive until day 21. Stool specimens were collected from the eight mice of each group (CAF2-1- and CAE3DU3-inoculated mice) on the indicated days. Calculated CFU/mL of the yeasts from stools are indicated for individual mice in the plots. The geometric mean is shown as a bar. Statistical analyses were performed using the Mann Whitney U test. Asterisks indicate significant differences (*** *p* < 0.001, ** *p* < 0.01).

**Figure 3 pathogens-10-00023-f003:**
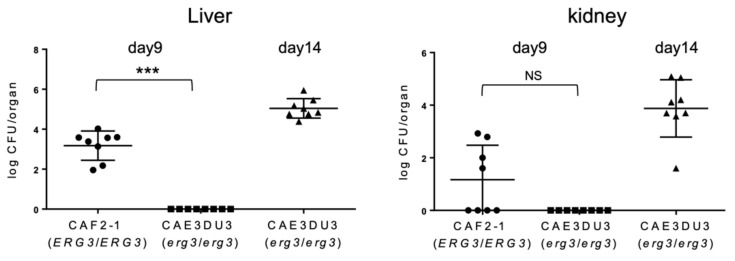
Comparison of the dissemination of yeast cells to the liver and kidneys. The livers and kidneys were removed from the eight mice of each group (CAF2-1- and CAE3DU3-inoculated mice) on the indicated days. Development of disseminated candidiasis was evaluated by calculating the log_10_ CFU/organ in the liver and kidney. Calculated CFUs of yeasts from the livers and kidneys are indicated for individual mice in the plots. The geometric mean is shown as a bar. Statistical analyses were performed using the Mann Whitney U test. Asterisks and NS indicate a statistically significant difference (*** *p* < 0.001) and no significance, respectively.

**Figure 4 pathogens-10-00023-f004:**
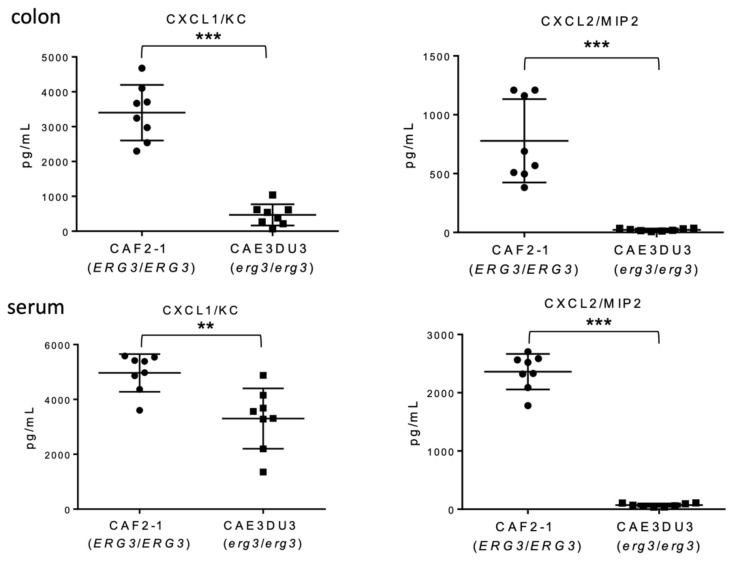
Comparison of chemokine induction between *C. albicans* strains CAF2-1 and CAE3DU3. Nine days after inoculation, the levels of CXCL1/KC and CXCL2/MIP-2 detected in the colon tissues and serum of CAE3DU3-inoculated mice were significantly lower than those detected in mice inoculated with CAF2-1. The geometric mean is shown as a bar. Statistical analyses were performed using the Mann Whitney U test. Asterisks indicate a statistically significant difference (*** *p* < 0.001, ** *p* < 0.01).

**Figure 5 pathogens-10-00023-f005:**
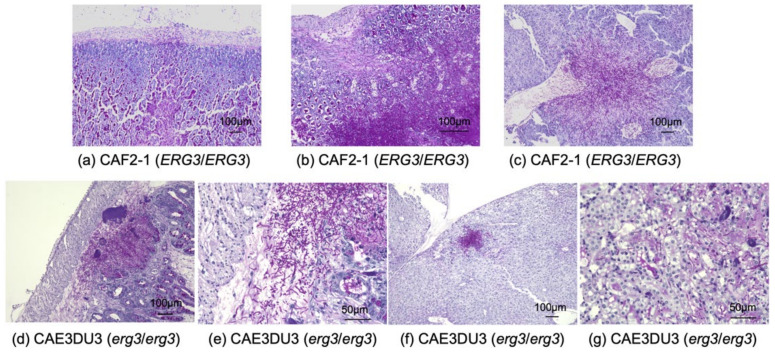
Histopathological examination of the organs obtained from mice infected with *C. albicans* strains. (**a**–**c**) Histopathological sections from mice infected with strain CAF2-1. Each organ was collected at 9 d post-inoculation. *C. albicans* was observed in both yeast and hyphal forms. *C. albicans* invaded the gut mucosa and liver tissues, and caused remarkable tissue necrosis; however, it was not detected in the kidneys. Sections of the (**a**) cecum, (**b**) colon, and (**c**) liver, at a magnification of ×100. (**d**–**g**) Histopathological sections from mice infected with strain CAE3DU3. Each organ was collected at 14 d post-inoculation. *C. albicans* was observed in both yeast and hyphal forms. *C. albicans* cells were found in the gut mucosa; however, the invasion and tissue necrosis was localized compared to that associated with CAF2-1. Cell accumulations were observed in the livers and kidneys without surrounding tissue necrosis. Section of (**d**) cecum at magnification ×100, (**e**) cecum at magnification ×400, (**f**) liver at magnification ×100, and (**g**) kidney at magnification ×400. Periodic acid Schiff staining was performed for all histopathological analyses.

## Data Availability

The data presented in this study are available on request from the corresponding author.

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
