# Peer review of "ERG3-Encoding Sterol C5,6-DESATURASE in Candida albicans Is Required for Virulence in an Enterically Infected Invasive Candidiasis Mouse Model"

_pathogens, 2020, doi:10.3390/pathogens10010023_

Round 1
Reviewer 1 Report
This is an important piece of work detailing the effect of erg3 mutations on virulence of Candida albicans in the mouse gut, which adds to our understanding of differing responses to these azole resistant mutants in different host tissues.
I have only a few comments for the authors;
The results section '2.1 Loss of ERG2 in C. albicans results in an enterically-infected invasion candidiasis mouse model' should include the fact that mice were treated with cyclophosphamide so it is clear why mortality is 100% in only a few days. The figure legend should also be re-labelled to include the term 'mouse model'.
In the phrase '....CXXL2/MIP-2 were lower in the colon tissue' in Figure 4, the word reduced should be replaced with lower.
Line 100, C. albicans should be Italicised.
Reviewer 2 Report
The manuscript entitled “ERG3 encoding sterol C5,6-desaturase in Candida albicans is required for virulence in an enterically-infected invasive candidiasis mouse model” describes the differential colonization, dissemination, and pathogenicity between a strain deleted for ERG3 gene and wild type strain of the fungal pathogen Candida albicans.
The manuscript is well written and well organized. My comments mostly reflect the potential need for clarification on the rational for some of the experimental set ups (in the form of additional text in results or discussion sections):
- Figure 1 and survival data
- The wild type strain kills animals in this model in a very reproducible manner it seems, with 50% mortality at day 9, 40% at day 10, and 10% at day 11. The window of operation with animal surviving the infection is tight. In comparison, the mutant strain seems to cause a delayed mortality at day 13 or 14. The inoculum chosen was presumably established from previous experiments and is required to result in mortality. Do the authors have data with smaller cell inoculum? and should the model ultimately be used for survival analyses? In this set up, colonization leads to mortality, yet C. albicans is a commensal organism.
- Following on that point, since animals infected with the wild type strain do not survive past 11 days, wouldn’t earlier collection time points have been insightful in understanding how invasive candidiasis develops over time, and could the authors comment on that in the discussion?
- Figure 2, and using animals not treated with antibiotic
- How does that figure relate to the other data since the treatment of the animal differs by the use or not of an antibiotic (to allow fungal colonization)? What is the rational for not using the same model with antibiotic but taking earlier time points (day 3, 5 and 9 for example).
- There seems to be one log difference between the 2 strains at all time points taken (Day 7, 14 and 21), but erg3 mutant is still present in the stools of infected mice, so it is not eliminated. Do the authors suspect that the mutant strain does not replicate as well as the wild type does in the gut environment?
- Figure 3 and taking samples at day 9 for both strains, and day 14 for the mutant strain group
- A suggestion could be to use only one graph per organ (wt and erg3 at day 9 + erg3 at day 14 on the same graph)
- Cell counts from the mutant strains at day 14 seem to be higher than for the wild type strain, which does not corroborate with the survival data in Figure 1 (in which 50% mortality is observed at day 9 for the animals infected with wild type, but 10 to 30% mortality is observed at day 14 for the animals infected with the mutant strain), do the authors have a potential explanation for this? In the discussion section (Lines 124-126), the authors state that “The mortality in mice infected with CAE3DU3 was thought to be a result of candidemia, since a large number of C. albicans cells were isolated from the liver and kidneys, and intestinal necrosis was not severe at 14 d post-infection.” But at the same time, results state (Lines 80-82) that no fungal cells were isolated in the blood samples at the time points taken. It would help to re-phrase the statement in the discussion statement to eliminate this discrepancy.
- Figure 4
- The data clearly highlight a major difference in the immune response between animals infected with wild type and those infected with the mutant strain. The data presented are from one time point only, day 9 post infection. Is there a rational for not showing data at day 14 for the animals infected with the mutant strains? It is mostly for consistency throughout the manuscript.
- Figure 5
- The text refers to a “milder” invasion and tissue necrosis (Lines 102 and in the figure legend). The method section does not include how quantification or qualification of the images was done. How was the milder invasion/necrosis quantified? Re-phrasing those statements may help if there was no quantification method used.
